# Health Risk Exposure Assessment of Migration of Perfluorooctane Sulfonate and Perfluorooctanoic Acid from Paper and Cardboard in Contact with Food under Temperature Variations

**DOI:** 10.3390/foods12091764

**Published:** 2023-04-24

**Authors:** Shu-Han You, Chun-Chieh Yu

**Affiliations:** Institute of Food Safety and Risk Management, National Taiwan Ocean University, Keelung City 20224, Taiwan

**Keywords:** perfluorooctane sulfonate, perfluorooctanoic acid, food-contact paper and cardboard, migration, health risk assessment

## Abstract

Perfluorooctane sulfonate (PFOS) and perfluorooctanoic acid (PFOA) are extensively used in food-contact paper and cardboard. However, they may migrate from food-contact materials to food, and the migration rate may be increased at elevated temperatures. In addition, there is a positive association of PFOS/PFOA levels with total cholesterol. Therefore, this study aims to assess the human health risk of increased total cholesterol associated with long-term exposure to PFOS and PFOA migration from food-contact paper and cardboard under temperature variation scenarios in adults. An exposure assessment was performed using an uptake dose model to estimate the uptake doses of PFOS and PFOA for the high-, intermediate-, and low-exposure scenarios. Benchmark dose (BMD) modeling was conducted to describe the dose–response relationships between PFOS/PFOA and total cholesterol levels. Finally, a margin of exposure (MOE) approach was used to characterize the risk. The results of the exposure assessment showed that PFOS and PFOA uptake doses in the high-exposure scenarios were around one and two orders of magnitude greater than those in the intermediate- and low-exposure scenarios, respectively. Under high-exposure scenarios, the uptake levels of hundredth-percentile PFOS and PFOA at high temperatures may raise health concerns (MOE < 1). This study provides a methodology to assess the health risks associated with exposure to migration of food contaminants from various types of paper and cardboard that come into contact with food.

## 1. Introduction

Perfluorooctane sulfonate (PFOS) and perfluorooctanoic acid (PFOA) are part of a group of artificial compounds called per- and polyfluoroalkyl substances. Due to their hydrophobic and lipophobic properties, PFOS and PFOA have been used extensively in water- and oil-proof products, such as fabrics, carpets, and food packaging [1,2]. They are bioaccumulative and do not easily degrade. In fact, they have been detected in humans, wildlife, and the environment [3]. In addition, the average half-lives of PFOS and PFOA in human serum are 5.4 and 2.3 years, respectively [4,5]. Exposure to PFOS and PFOA has been studied in rodents for toxicological endpoints, such as reproduction and development, pulmonary, neurological, and liver toxicities [6,7,8,9,10,11,12,13,14].

PFSA exposure based on epidemiological studies in humans have many potential adverse health effects, including altered immune and thyroid function, liver disease, lipid and insulin dysregulation, kidney disease, adverse reproductive and developmental outcomes, and cancer [15]. There are also associations between PFAS exposure and lipid level among populations. Zeng et al. [16] indicated that PFOS, PFOA, and PFNA were positively associated with total cholesterol, low-density lipoprotein, and triglycerides in Taiwanese children. Lin et al. [17] have found that PFOS exposure is significantly positively-correlated with platelet distribution width, mean platelet volume, and platelet–large cell ratio.

Food-contact materials are potential sources of oral exposure to PFOS and PFOA [18,19]. Since PFOS and PFOA possess oil-proof properties, they may be used in the processing of food-contact paper and cardboard. Non-stick cookware and food-contact paper and cardboard are the most recognizable food-contact materials. There is a significantly higher migration rate of perfluorinated compounds (PFCs) into food from food-contact paper and cardboard compared to non-stick cookware [19]. Different concentrations of PFOS and PFOA have been detected in various types of food-contact paper and cardboard. They are mostly found in fast food packaging and popcorn bags [20,21,22,23,24,25,26,27,28,29].

PFOS and PFOA as food contaminants have the potential to migrate from paper and cardboard in contact with food [30]. The rate of migration depends on various factors, including physicochemical properties, time, and temperature [31]. Due to the complex physicochemical properties of the food matrix and food-contact paper and cardboard, real food is often replaced by food simulants during migration tests. Previous studies conducted migration tests of PFCs from food-contact paper and observed that the rate of migration (in percentage) increased with time and temperature [18,30,32]. When examining the migration of PFOS and PFOA from food-contact paper and cardboard into food, temperature may be the most significant factor due to the brief interaction time between food-contact paper and cardboard and food.

Trudel et al. [33] applied a scenario-based approach to estimate consumer exposure to PFOS and PFOA. The high-exposure scenarios for teenagers and adults were dominated by populations with exposure to PFOA from food-contact paper and cardboard. One of the limitations of the study was uncertainty about the input parameters, such as migration rates and the concentration of contaminants in food-contact paper and cardboard. To better understand food safety limitations, PFOS and PFOA assessments should more accurately specify the range of input parameter values. Most of the recent studies on PFCs in food-contact paper and cardboard have focused on exposure assessment, including the development and optimization of analysis methods and migration tests [18,23,28,30,32]. These studies indicate that PFOS and PFOA are still present in food-contact paper and cardboard.

To our knowledge, no study has evaluated the health risks associated with PFOS and PFOA found in food-contact paper and cardboard in Taiwan. This study aims to assess the risks to human health based on an increase in total cholesterol associated with long-term exposure to migration of PFOS and PFOA from food-contact paper and cardboard under temperature variations in adults.

Only one study has collected food-contact paper in Taiwan to conduct PFOA migration tests and an initial risk assessment [34]. However, the sample size was insufficient to represent the migration conditions of PFOA in food-contact paper in general.

## 2. Materials and Methods

### 2.1. Study Framework

Figure 1 illustrates the study framework included in the risk assessment. First, the formulation of the problem showed that PFOS and PFOA could migrate from food-contact paper and cardboard into food under temperature variations (Figure 1A). Then, human exposure to the food contaminated by PFOS and PFOA may result in a positive association between concentrations of total cholesterol and PFOS/PFOA in human blood (Figure 1A).

Second, the exposure assessment was based on a published experimental dataset and estimated input parameters, and the uptake doses of PFOS and PFOA from food-contact paper and cardboard were estimated using the uptake dose model (Figure 1B). PFOS and PFOA concentrations (internal doses) were converted to dietary intake (external doses) using a one-compartment steady-state pharmacokinetic (PK) model (Figure 1B). Third, based on the epidemiological data after dose conversion, the evaluation of the dose–response assessment for PFOS/PFOA and total cholesterol in human blood was performed by benchmark dose (BMD) modeling (Figure 1B). Finally, for the risk characterization, we used a margin of exposure (MOE) approach to integrate exposure and dose–response assessments. The MOE depends on the idea of “without zero” (acceptable level of risk) for plotting risk estimates (Embry et al. [34]). This is used to compare the exposure and toxicity estimates on the RISK21 matrix.

### 2.2. Study Data

We collected data on the levels of PFOS and PFOA in food-contact paper and cardboard published during 2007–2017 [20,21,22,23,24,25,26,27,28,29]. The criteria for selecting papers were the PFOS/PFOA levels measured by liquid chromatography–mass spectrometry with recovery rate greater than 80%. Briefly, the levels of PFOS and PFOA were analyzed via liquid chromatography–tandem mass spectrometry (LC/MS or LC/MS/MS), and the features of the analysis methods are shown in Appendix A. Six types of food-contact paper and cardboard were included in this study: popcorn bags, paper tableware, paper boxes, paper cups, paper bags, and wrappers. Considering the normalized thickness data for all categories, the mass units of PFOS and PFOA concentrations in each category were all converted to area units (ng/cm^2^) [23] (Appendix A). For a conservative evaluation, the values of PFOS and PFOA concentrations between the limit of quantification (LOQ) and the limit of detection (LOD) were assigned as half of the LOQ [34]. The undetected values and the values less then the LOD were assigned as half of the LOD (if values were not reported, LOD and LOQ for published data were assigned as 0) [34].

The migration rates of PFOS and PFOA under temperature variations were obtained from the migration tests conducted by Elizalde et al. [32] and Xu et al. [30]. According to guidance for food-contact materials [2,35], we set two temperature scenarios: a high-temperature scenario (100–120 °C), and a low-temperature scenario (40 °C). Then, we used TableCurve 2D (Version 5.01; AISN Software, Mapleton, OR, USA) to perform the curve-fitting with migration rates.

We collected samples for popcorn bags (N = 3), paper tableware (N = 3), paper boxes (N = 3), paper cups (N = 3), paper bags (N = 3), and wrappers (N = 3) from local convenience stores and fast food restaurants in Keelung City. After washing and cutting the samples, they were stored at room temperature to measure the area. Considering the irregular surface of food, we assumed 80% of the measured area as the contact area.

We adopted the experimental data from Hundley et al. [36] to calculate the uptake fractions of PFOS and PFOA (Appendix A). Briefly, the experimental design in which Hundley et al. [36] treated male and female mice, rats, hamsters, and rabbits with a single 10 mg kg^−1^ oral dose of ^14^C-ammonium perfluorooctanoate. The ^14^C radioactivity and content by sex and species were analyzed by liquid scintillation spectroscopy to determine excretion and tissue distributions. The percent dose ranged from 66% to 92.9% in male and female mice, rats, hamsters, and rabbits.

To examine the effect of PFOS/PFOA on body weight, adult weights were obtained from the Health Promotion Administration, Ministry of Health and Welfare (MOHW, 2019). The values of the market fraction of PFC-treated paper and cardboard, the frequency of food contact with treated paper and cardboard, and the duration of contact were obtained from Trudel et al. [33].

We collected the epidemiological data on the relationships between PFOS/PFOA and total cholesterol in blood from Eriksen et al. [37], Nelson et al. [38], and Steenland et al. [39] to conduct the dose–response assessment. We also obtained the serum concentrations of PFOS and PFOA in Taiwanese adults from Hsu et al. [40] to estimate their PFOS and PFOA dietary intake (Appendix A).

### 2.3. Exposure Assessment

An uptake dose model was used to estimate the exposure to PFOS and PFOA from paper and cardboard in contact with food that varied with temperature variations in adults. Uptake doses were calculated at high (100–120 °C) and low (40 °C) temperatures for the high- (95%), intermediate- (50%), and low-exposure (5%) scenarios. The uptake doses of PFOS and PFOA migration from food-contact paper and cardboard (*D*_fcp_) (ng kg-bw^−1^ d^−1^) were estimated by the following uptake dose model (Trudel et al. [33]):(1)$$D_{fcp}=\frac{C_{pb}\cdot r_{migr}\cdot {MF}_{pb}\cdot f_{food\_pb}\cdot A_{c}\cdot t_{c}}{m_{bw}}\cdot F_{uptake}$$
where *C*_pb_ is the concentration of PFOS and PFOA in food-contact paper and cardboard (ng cm^−2^), *r*_migr_ is the migration rate of PFOS and PFOA from food-contact paper and cardboard into the food (h^−1^), *MF*_pb_ is the market fraction of PFC-treated paper and cardboard (%), *f*_food_pb_ is the frequency of food contact with treated paper and cardboard (d^−1^), *A*_c_ is the contact area of food-contact paper and cardboard with food (cm^2^), *t*_c_ is the duration of contact (h), *F*_uptake_ is the uptake fraction of PFOS and PFOA (%), and *m*_bw_ is the body weight for male and female adults (kg-bw). All parameters were imported into Crystal Ball^®^ (Version 2000.2, Decisioneering, Inc., Denver, CO, USA) to estimate the PFOS and PFOA uptake doses. Crystal Ball^®^ was employed to implement Monte Carlo simulation, which performed 10,000 iterations to ensure the uncertainty of the simulation results.

### 2.4. Dose–Response Assessment

A PK model was used to assess the dose–response relationship between the intake of PFOS/PFOA (*Dose*) and total cholesterol levels (*Response*). In addition, the model was used to estimate doses (PFOS and PFOA dietary intake) and convert serum/plasma concentrations of PFOS/PFOA to dietary intake. The total cholesterol concentrations were adopted from Eriksen et al. [37], Nelson et al. [38], Steenland et al. [39], and a study conducted by Hsu et al. [40] in Taiwan (Appendix A). Based on an assumption of steady-state conditions, the PK model was expressed as
(2)DP=CP×KP×Vd
where *DP* is the dietary intake of PFOS and PFOA (ng kg-bw^−1^ d^−1^), *CP* represents the PFOS/PFOA concentration in serum or plasma (ng mL^−1^) [41,42,43,44], *KP* is the elimination rate of PFOS/PFOA (d^−1^) ($KP=ln2/T_{\frac{1}{2}}$), $T_{\frac{1}{2}}$ denotes the half-life of PFOS/PFOA in human serum (d), and *Vd* is the distribution volume of PFOS/PFOA (mL kg-bw^−1^). The half-lives and distribution volumes of PFOS and PFOA were adopted from Bartell et al. [4], Harada et al. [45], and Olsen et al. [5], and they are presented in Appendix A.

Then, we used Benchmark Dose Software (U.S. EPA’s Benchmark Dose Software version 3.1.1) to conduct BMD modeling and calculate BMDL_5_. We set the benchmark response (BMR) to a 5% relative deviation in total cholesterol. We also followed the recommended six analysis steps for U.S. EPA’s Benchmark Dose Software (Davis et al. [46]). The total cholesterol concentrations [37,38,39] were imported into Benchmark Dose to fit the dose–response models.

### 2.5. Risk Characterization

We used two approaches to integrate the results of the exposure and dose–response assessments to conduct the risk characterization. One was the MOE approach, which involves dividing the toxicity data by the estimated human intake. If MOE < 1, then there is a potential health risk. If MOE > 1, then there is no health concern. The RISK21 matrix is a visualization method for describing the MOE (Embry et al. [47]). We used the RISK21^®^ Webtool (Version 2.0) to conduct the MOE approach. Exposure and toxicity data (5–100th-percentile uptake doses of PFOS and PFOA, and 95% confidence limit BMDs) were plotted on the *X*- and *Y*-axes of the RISK21 matrix, respectively.

The other approach was to estimate the percentages of uptake doses of PFOS and PFOAs from paper and cardboard in contact with food for Taiwanese adults. The medians and 95th-percentiles of uptake doses of PFOS and PFOA for the high-exposure scenario were compared to those of PFOS and PFOA dietary intake (converted by the PK model) of Taiwanese adults.

## 3. Results and Discussion

### 3.1. Input Parameters

The input parameters of the uptake dose model for the different scenarios are summarized in Table 1. As shown in Figure 2, the median concentrations of PFOS and PFOA in the six categories were lower than 0.15 ng/cm^2^. The highest concentrations of PFOS and PFOA were 0.61 ng/cm^2^ and 1.76 ng/cm^2^, observed in the paper box (Figure 2A) and the popcorn bag (Figure 2B), respectively. The distributions of PFOS concentrations in food-contact paper and cardboard were estimated using a lognormal (LN) function with a geometric mean (GM) of 0.02 ng/cm^2^ and a geometric standard deviation (GSD) of 6.45, denoted as LN (0.02, 6.45) (Appendix A). The distributions of PFOA concentrations in all categories were estimated using an LN function, denoted as LN (0.016, 6.38) (Appendix A). The concentration ranges (minimum–maximum) of PFOS and PFOA were 0.002–0.398 ng/cm^2^ and 0.004–0.796 ng/cm^2^, respectively (Appendix A).

Figure 3A,B shows the fitting models of PFOA migration (%) at high temperatures (100–120 °C) for 125 min and at low temperature (40 °C) for 250 h. Migration (%) was estimated using the following models:(3)$$Migration\left(\%\right)=2.37+1.6{time}^{0.5}$$(4)$$Migration(\%)=4.95+2.93{time}^{0.5}$$

The adjusted *r^2^* values of the models were 0.78 and 0.93, respectively. The migration rates at high and low temperatures were 0.15 h^−1^ and 0.08 h^−1^, respectively (Figure 3C). Since there were less data on PFOS migration from food-contact paper and cardboard, we assumed the migration rates of PFOS and PFOA to be the same. However, this may have resulted in uncertainty in the PFOS and PFOA exposure assessment.

The average contact areas of the different types of food-contact paper and cardboard are shown in Appendix A. The distributions of the contact area of food-contact paper and cardboard with food were estimated using a normal (N) function with a mean (M) of 510.05 cm^2^ and a standard deviation (SD) of 256.62, denoted as N (510.05, 256.62) (Appendix A). The distributions of the uptake fractions of PFOS and PFOA, male body weight, and female body weight were estimated using an N function with the Ms and SDs, denoted as N (0.82, 0.1), N (70.4, 19.79), and N (58.0, 17.51), respectively (Appendix A).

### 3.2. Uptake Doses under Exposure Scenarios

The distributions of PFOS and PFOA uptake doses in males and females from paper and cardboard in contact with food at high and low temperatures in the high-, intermediate-, and low-exposure scenarios are shown in Appendix A. The uptake doses of PFOS in the high-exposure scenario were approximately one and three orders of magnitude higher than those in the intermediate- and low-exposure scenarios, respectively (Figure 4). PFOS uptake doses at high temperatures were two times greater than doses at low temperature in all exposure scenarios (Figure 4). PFOA uptake doses were similar to PFOS in the different scenarios (Figure 5).

The uptake doses of 100th-, 95th-, 50th-, and 5th-percentile PFOS/PFOA from paper and cardboard in contact with food at high and low temperatures in the high-, intermediate-, and low-exposure scenarios are shown in Appendix A. Estimated 100th-percentile PFOA uptake doses were greater than 100th-percentile PFOS uptake doses because there were more published high-concentration data for PFOA in food-contact paper (Appendix A).

Appendix A shows that the median PFOS uptake doses for males at high and low temperatures were 1.7 × 10^−2^ (5–95th-percentile: 7.8 × 10^−4^–3.2 × 10^−1^) and 9.3 × 10^−3^ (4.2 × 10^−4^–1.7 × 10^−1^) ng kg-bw^−1^ d^−1^, respectively. The Median PFOS uptake doses for females at high and low temperatures were 2.0 × 10^−2^ (9.4 × 10^−4^–3.7 × 10^−1^) and 1.1×10^−2^ (5.0 × 10^−4^–2.0 × 10^−1^) ng kg-bw^−1^ d^−1^, respectively. PFOA uptake doses for males at high and low temperatures were 1.6 × 10^−2^ (6.2 × 10^−4^–3.4 × 10^−1^) and 8.3 × 10^−3^ (3.3 × 10^−4^–1.8 × 10^−1^) ng kg-bw^−1^ d^−1^, respectively. PFOA uptake doses for females at high and low temperatures were 1.8 × 10^−2^ (7.5 × 10^−4^–4.1 × 10^−1^) and 9.8 × 10^−3^ (4.0 × 10^−4^–2.2 × 10^−1^) ng kg-bw^−1^ d^−1^, respectively. Results showed that the uptake doses of PFOS and PFOA for females were slightly greater than those in males because the *m_bw_* of females was lower than that of males. We used the same *f*_food_pb_ to estimate their PFOS and PFOA uptake doses (Figure 4 and Figure 5).

### 3.3. BMDL5 of PFOS and PFOA

The dose–response relationships of PFOS/PFOA intake (Dose) with total cholesterol concentrations (Response) are shown in Figure 6. The models with a BMR of 5% relative deviation for the BMDs of PFOS and PFOA and a 95% lower confidence limit for the BMDL_5_s were expressed as
(5)$$Response=195.35\times \left[1.07-\left(1.07-1\right)\times exp\left(-{\left(0.46\times Dose\right)}^{1.8}\right)\right]$$
(6)$$Response=192.78\times \left[1.07-\left(1.07-1\right)\times exp\left(-0.42\times Dose\right)\right]$$

The BMDL_5_ values for PFOS and PFOA were 1.67 ng kg-bw^−1^ d^−1^ and 1.46 ng kg-bw^−1^ d^−1^, respectively (Figure 6). The BMDL_5_ values of PFOS and PFOA derived in our study were close to the BMDL_5_ (PFOS: 1.86 ng kg-bw^−1^ d^−1^; PFOA: 0.86 ng kg-bw^−1^ d^−1^) derived from EFSA (2018). The 95% confidence limit BMD values for PFOS and PFOA were 1.67–2.7 ng kg-bw^−1^ d^−1^ and 1.46–4.8 ng kg-bw^−1^ d^−1^, respectively (Figure 6).

### 3.4. Health Risk Assessment of Increased Total Cholesterol

The ranges for exposure estimates of PFOS and PFOA (5–100th-percentile uptake doses) and toxicity (95% confidence limit BMDs) were plotted on the RISK21 matrix. The shapes and positions of the intersecting areas were used to visualize the total cholesterol risks for the decision-making process. The results at high and low temperatures in the high-exposure scenario are shown for each sex (Figure 7 and Figure 8).

The exposure-/toxicity-intersecting areas for the 5–95th-percentile PFOS and PFOA uptake doses were on the left side of the MOE line segment (MOE > 1), which indicated that the exposure levels were below the BMDL_5_ for toxicity (Figure 7 and Figure 8). However, some of the exposure-/toxicity-intersecting areas for the 95–100th-percentile of uptake doses of PFOS and PFOA at high temperatures were on the right side of the MOE line segment (MOE < 1), which indicated the possibility of health risks (Figure 7 and Figure 8). Higher concentrations of PFOA in food-contact paper and cardboard were used in the data collection, so the exposure/toxicity intersecting areas for the 95–100th-percentile PFOA uptake doses at low temperature were on the right side of the MOE line segment (Figure 8B,D).

### 3.5. Percentages in Dietary Intakes

The PFOS and PFOA dietary intake of Taiwanese adults is shown in Appendix A. The PFOS dietary intakes of males and females were 0.49–2.39 ng kg-bw^−1^ d^−1^ (median: 1.08) and 0.36–2.71 ng kg-bw^−1^ d^−1^ (0.63), respectively (Appendix A). The PFOA dietary intakes of males and females were 0.51–1.90 ng kg-bw^−1^ d^−1^ (median: 0.93) and 0.38–1.48 ng kg-bw^−1^ d^−1^ (0.61), respectively (Appendix A). Chen et al. [49] analyzed and estimated the dietary intake of the general Taiwanese population based on the concentrations of PFCs in food, finding that PFOS and PFOA dietary intakes were 0.46 ng kg-bw^−1^ d^−1^ and 85.1 ng kg-bw^−1^ d^−1^, respectively. The PFOA dietary intake in Chen et al. [49] was significantly higher than that in our study, which may have been due to the higher levels of PFOA contamination in the analyzed food.

Figure 9 shows the percentage contributions of PFOS and PFOA uptake doses from food-contact paper and cardboard to adult dietary intake at high and low temperatures in the high-exposure scenario. The contributions of median PFOS and PFOA uptake doses to both male and female dietary intake were less than 4% (Figure 9). The contributions of 95th-percentile PFOS uptake doses at high and low temperatures to male and female dietary intake were 29% and 16%, and 58% and 31%, respectively (Figure 9A). The contributions of 95th-percentile PFOA uptake doses at high and low temperatures to male and female dietary intake were 37% and 20%, and 67% and 36%, respectively (Figure 9B). The percentages of uptake doses of PFOS and PFOA for females were higher than those in males because females had lower dietary intakes of both PFOS and PFOA. However, uncertainties may occur when PFOS and PFOA concentrations and migration rates vary between different types of food-contact paper and cardboard. The uncertainty of source contribution to the concentration and migration rate may result from the thickness of the layer, the conversion of units of measurement, brand, and manufacturing country for food-contact paper and board.

## 4. Conclusions

This study is the first systematic health risk assessment of the migration of PFOS and PFOA from food-contact paper and cardboard conducted in Taiwan. The estimated median concentrations of PFOS and PFOA from food-contact paper and cardboard were all lower than 0.11 ng cm^−2^. PFOS and PFOA uptake doses at high temperatures were two times greater than doses at low temperature. The BMDL_5_ values for PFOS and PFOA were 1.67 and 1.46 ng kg-bw^−1^ d^−1^, respectively. Based on MOE, PFOS and PFOA from food-contact paper and cardboard does not pose any health risks (MOE > 1). However, the 100th-percentile uptake doses of PFOS and PFOA at high temperatures in high-exposure scenarios may cause health concerns (MOE < 1). This study provides an approach for conducting health risk assessments of exposure to migration of food contaminants from various types of food-contact paper and cardboard. Future research should collect more detailed dietary exposure data for the risk assessment of different types of food-contact paper and cardboard.

## Figures and Tables

**Figure 1 foods-12-01764-f001:**
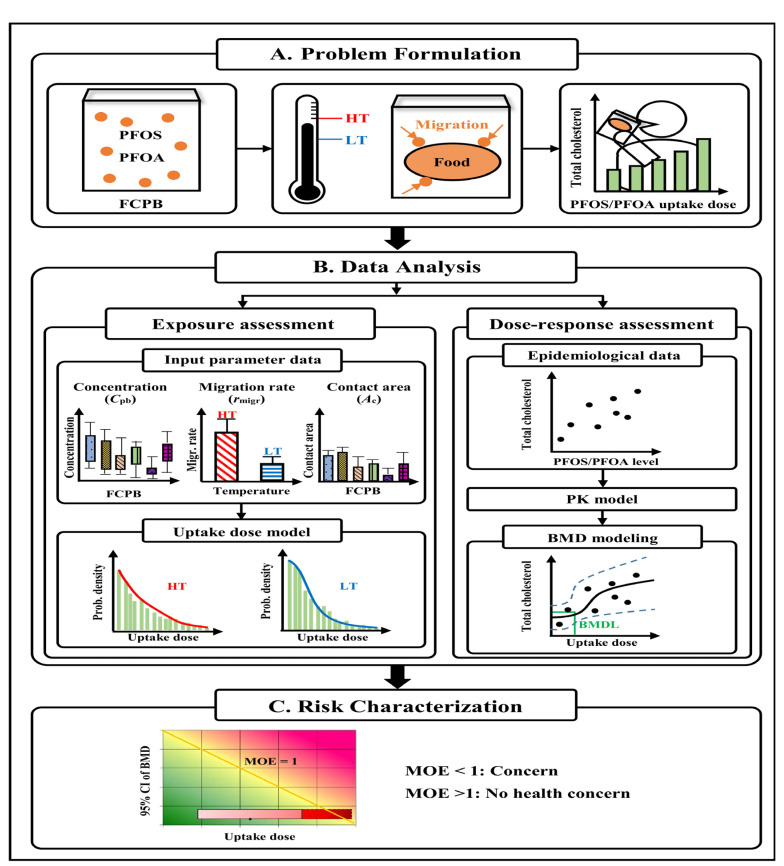
Study framework.

**Figure 2 foods-12-01764-f002:**
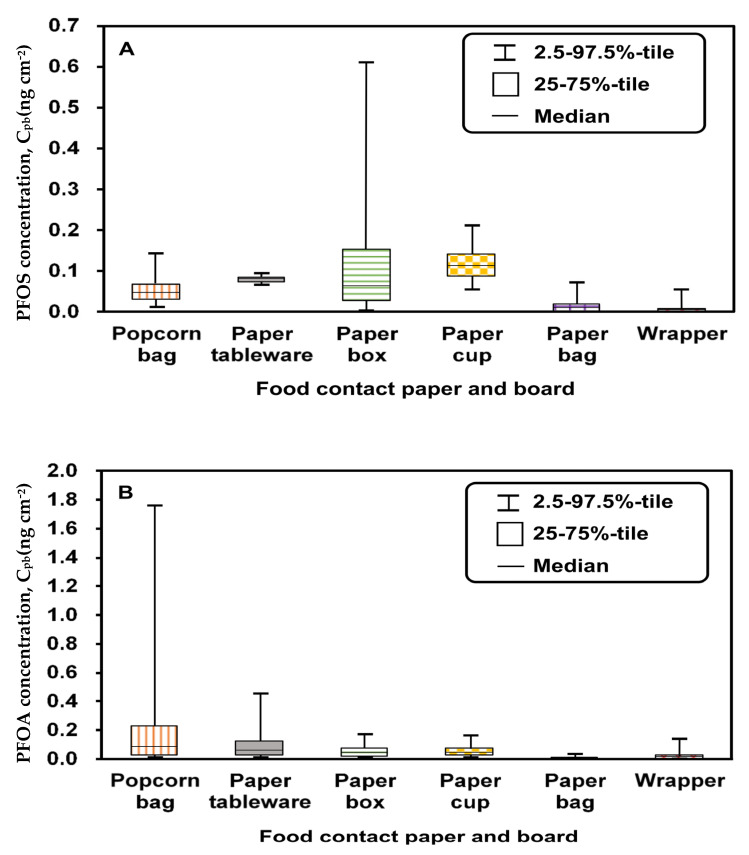
Concentrations of (**A**) PFOS and (**B**) PFOA in different types of food-contact paper and board (without concentrations lower than the limit of quantification).

**Figure 3 foods-12-01764-f003:**
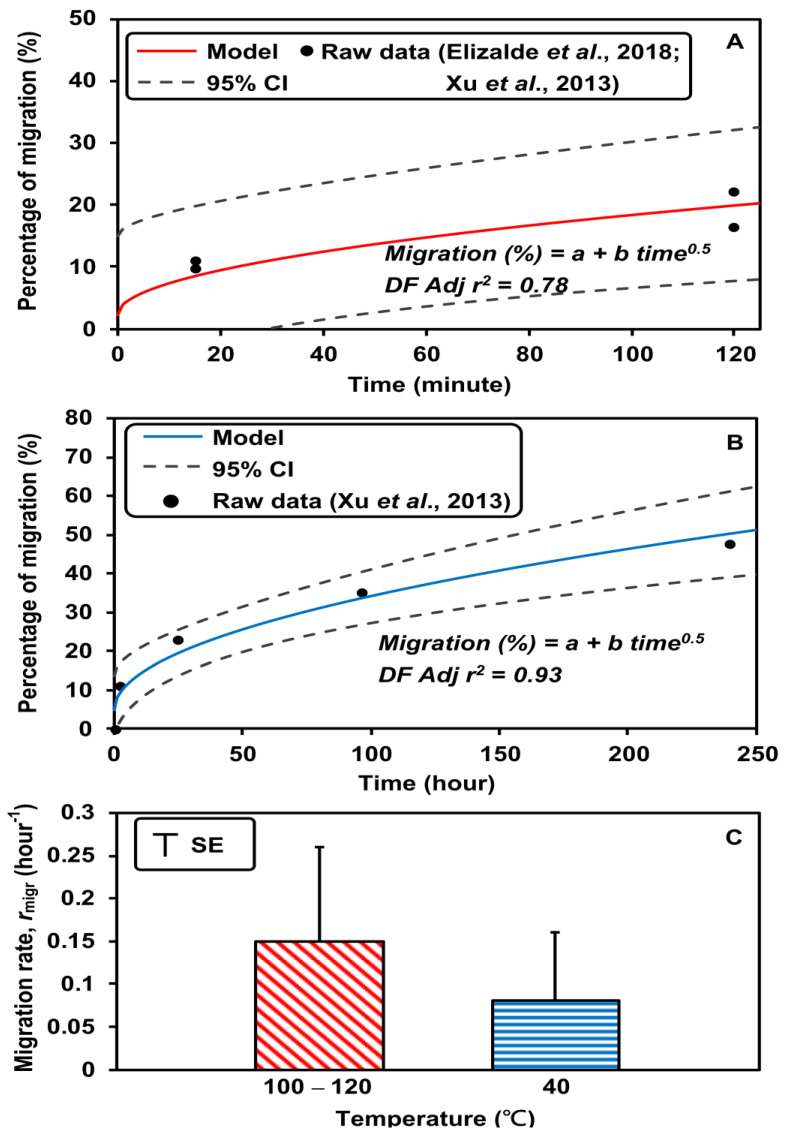
Percentages of PFOA migration from food-contact paper and board at (**A**) high (100–120 °C) and (**B**) low (40 °C) temperatures, and (**C**) migration rates at high and low temperatures (Confidence interval (CI); Degree of freedom adjust *r^2^* (DF Adj *r^2^*)). Raw data (Elizalde et al. [32] and Xu et al. [30]).

**Figure 4 foods-12-01764-f004:**
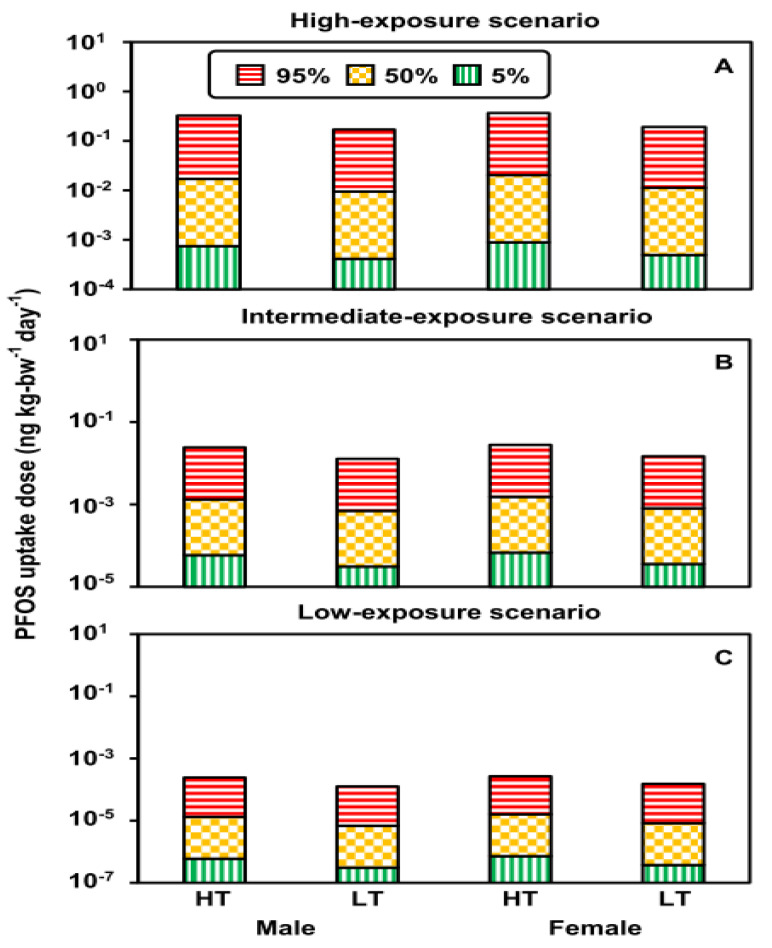
The 95th-, 50th- and 5th-percentile PFOS uptake doses for adults in the different exposure scenarios ((**A**) high temperature (HT): 100–120 °C; (**B**) low temperature (LT): 40 °C). (**C**) The 95th-, 50th- and 5th-percentile PFOS uptake doses for sexes in the different exposure scenarios.

**Figure 5 foods-12-01764-f005:**
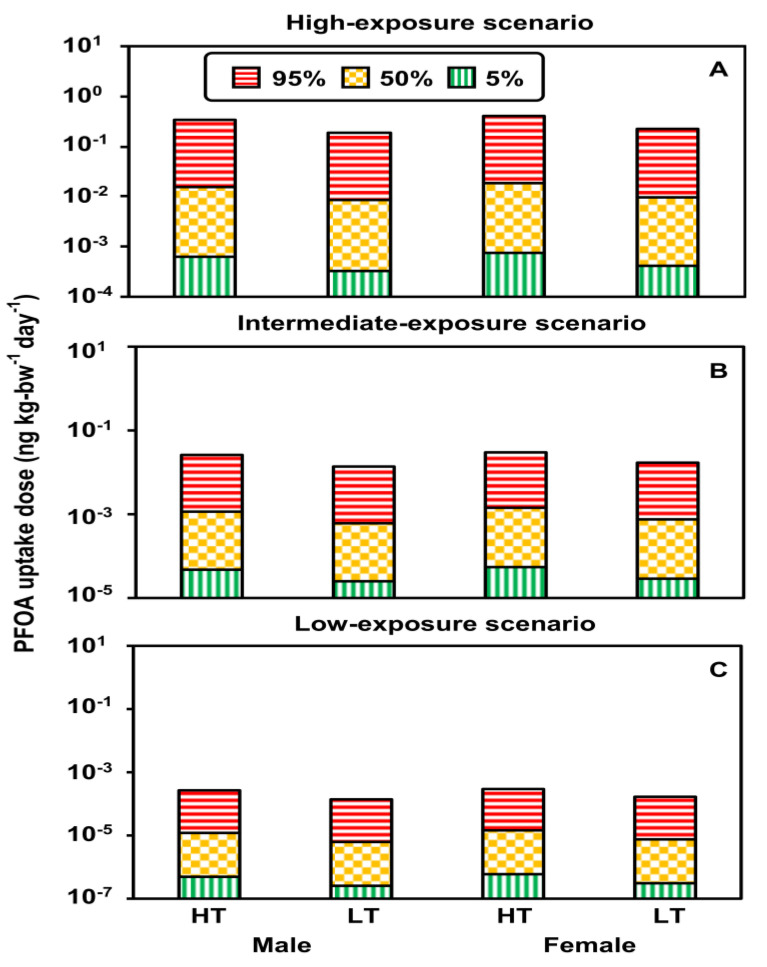
The 95th-, 50th- and 5th-percentile PFOA uptake doses for adults in the different exposure scenarios ((**A**) high temperature (HT): 100–120 °C; (**B**) low temperature (LT): 40 °C). (**C**) The 95th-, 50th- and 5th-percentile PFOA uptake doses for sexes in the different exposure scenarios.

**Figure 6 foods-12-01764-f006:**
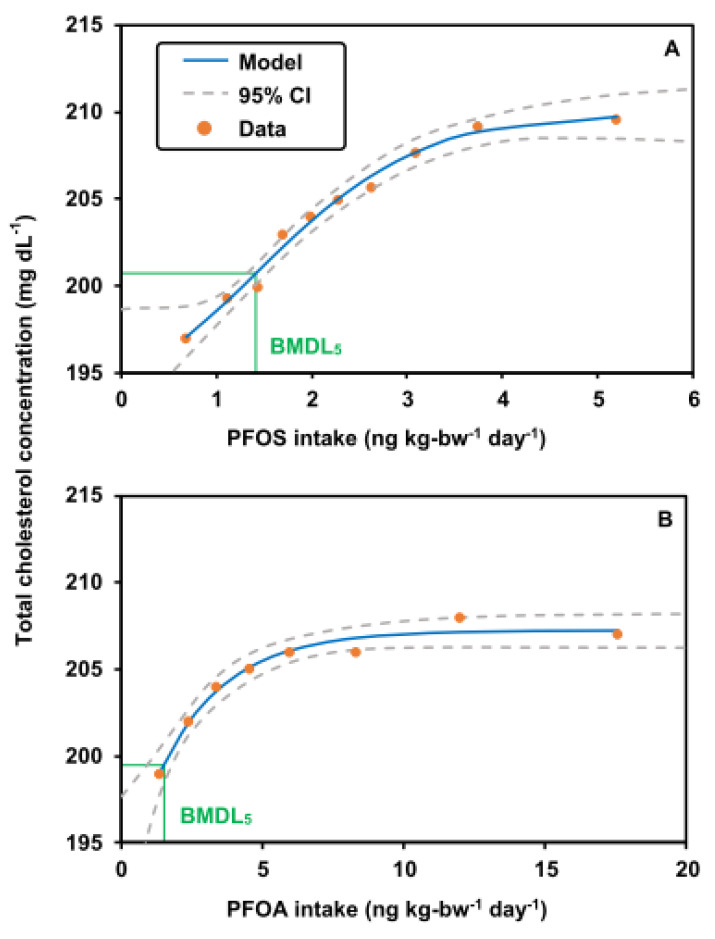
Dose–response relationships between (**A**) PFOS or (**B**) PFOA concentrations and total cholesterol concentrations (confidence interval (CI)).

**Figure 7 foods-12-01764-f007:**
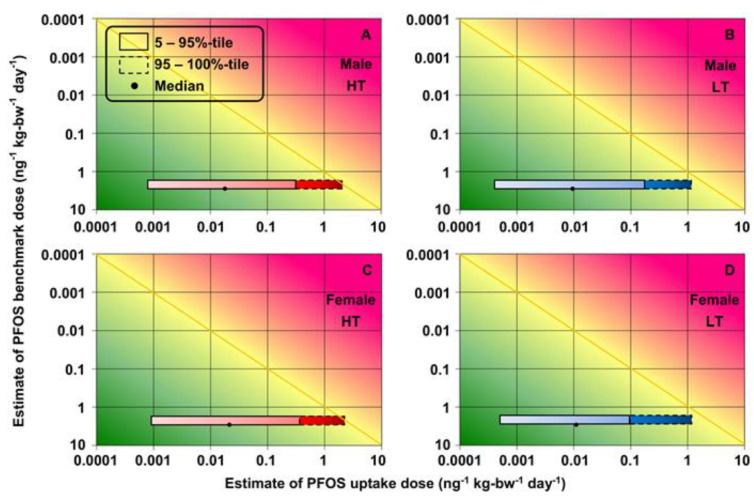
Risks of increased total cholesterol associated with exposure to PFOS migration of food-contact paper and board for (**A**,**B**) males and (**C**,**D**) females at high and low temperatures in high-exposure scenario, respectively (high temperature (HT): 100–120 °C; low temperature (LT): 40 °C).

**Figure 8 foods-12-01764-f008:**
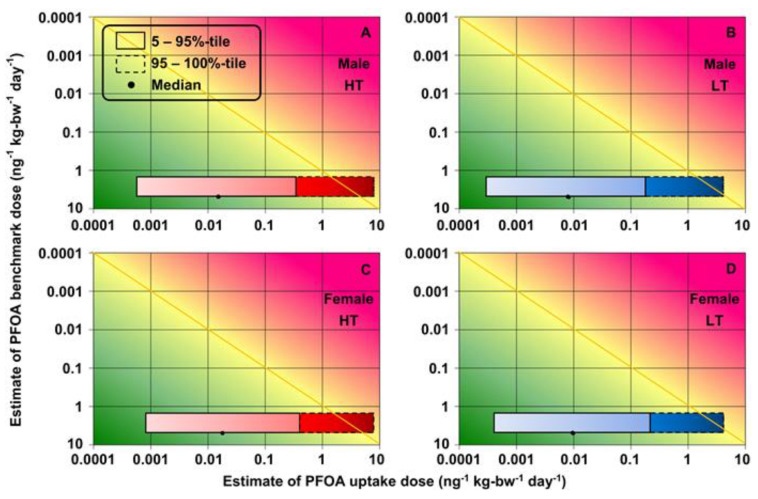
Risks of increased total cholesterol associated with exposure to PFOA migration of food-contact paper and board for (**A**,**B**) males and (**C**,**D**) females at high and low temperatures in high-exposure scenario, respectively (high temperature (HT): 100–120 °C; low temperature (LT): 40 °C).

**Figure 9 foods-12-01764-f009:**
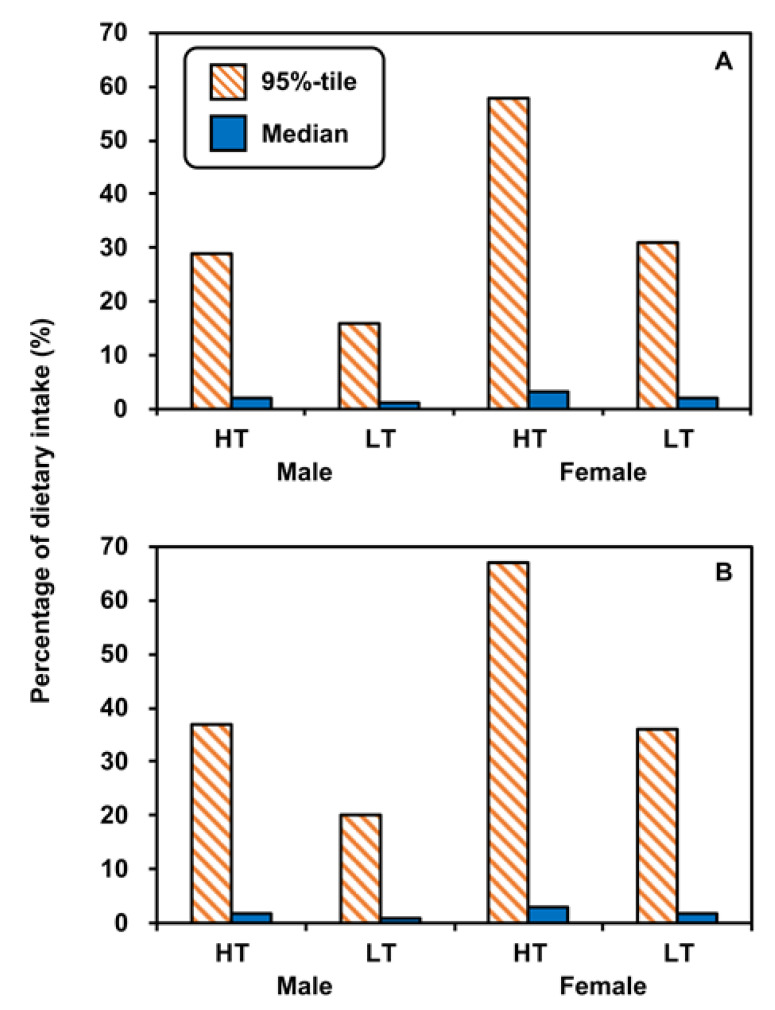
Contributions of (**A**) PFOS and (**B**) PFOA uptake doses to the dietary intakes of adults in the high-exposure scenario (high temperature (HT): 100–120 °C; low temperature (LT): 40 °C).

**Table 1 foods-12-01764-t001:** Input parameters of the uptake dose model for the different scenarios.

		Exposure Scenario			
Parameter	Unit	Low	Intermediate	High	Reference
*C* _pb_	ng cm^−2^	Collecting data^a^			This study
*r* _migr_	hour^−1^	HT: 0.15 (100–120 °C) ^b^ LT: 0.08 (40 °C) ^b^			This study
*MF* _pb_	%	10	50	100	Trudel et al. [33]
*f* _food_pb_	day^−1^	0.03	0.3	1	Trudel et al. [33]
*A* _c_	cm^2^	Measuring ^c^			This study
*t* _c_	hour	0.25	0.5	1	Trudel et al. [33]
*F* _uptake_	%	Simulating ^d^			This study
*m* _bw_	kg-bw	70.4 ±19.79 ^e^ (Male) 58.0±17.51 ^e^ (Female)			MOHW [48]

Abbreviations: *C*_pb_ is the concentration of PFOS and PFOA in food-contact paper and board; *r*_migr_ is the migration rate of PFOS and PFOA from food-contact paper and board into the food; *MF*_pb_ is the market fraction of perfluorinated compound-treated paper and board; *f*_food_pb_ is the frequency of contact of food with treated paper and board; *A*_c_ is the area of food-contact paper and board in contact with food; *t*_c_ is the duration of contact; *F*_uptake_ is the uptake fraction of PFOS and PFOA; *m*_bw_ is the body weight for male and female adults; HT is high temperature; LT is low temperature. ^a^ *C*_pb_ data in food-contact paper and board were collected from 10 published papers from 2007–2017 [20,21,22,23,24,25,26,27,28,29]. ^b^ *r*_migr_ were calculated according to Xu et al. [30] and Elizalde et al. [32]. ^c^ *A*_c_ were measured from food-contact paper and board purchased from a local market. ^d^ *F*_uptake_ were simulated according to Hundley et al. [36]. ^e^ Mean ± SD.

## Data Availability

The data can be obtained from the corresponding author and are presented in this manuscript.

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
