# Peer review of "Health Risk Exposure Assessment of Migration of Perfluorooctane Sulfonate and Perfluorooctanoic Acid from Paper and Cardboard in Contact with Food under Temperature Variations"

_foods, 2023, doi:10.3390/foods12091764_

Round 1

Reviewer 1 Report

This is an excellent paper, very useful as a complete guideline for this type of risk exposure assessment. The readability of this MS is good and the experimental design has been well described together with a complete description of the obtained results.

Minor revision needed:

The food safety risk related to PFOA and PFOS intake is not only related to cholesterol increase (please see for example: Tsai et al (2020) A case-control study of perfluoroalkyl substances and the risk of breast cancer in Taiwanese women. Environment International, 142, 105850. Doi: 10.1016/j.envint.2020.105850). In the introduction, please add more comments related to the overall food safety risk linked to these substances.

The Margin of Exposure approach usually takes as reference 10000 as the minimum threshold for substances both genotoxic and carcinogenic (EFSA (2012) Scientific Opinion on the applicability of the Margin of Exposure approach for the safety assessment of impurities which are both genotoxic and carcinogenic in substances added to food/feed. EFSA Journal, 10(3), 2578. doi:10.2903/j.efsa.2012.2578). In this study, the authors refer to a threshold of 1 which is usually considered when working with the “Margin of Safety” (MoS) approach. Please revise the definitions and add more explanation about the approach used in this regard, since the most significant findings of this study are strictly related to these indexes.

Title: a slight modification is suggested: Health risk exposure assessment to migration of perfluorooctane…..

Abstract: please improve English at lines 11-12. Line 13: risk

Please report the references in the text according to the Journal’s style.

Line 53: please improve readability.

Lines 104-105: please adds more info about criteria used for selecting papers.

Lines 115-118: Not clear. Values without the LOD? Please explain better about this section.

Line 125: please add as many info as possible about these samples.

Lines 132-134: this section seems overlooked. Please add more info.

Please check and the standardize the font used for “°C” throughout the paper and in tables/figures.

The resolution of text in figures should be improved.

Line 194: please check a keying error.

Figure 2 caption: please specify the percentage of values higher than the LOQ (and the LOQ itself) in the caption or in the text.

Figures: please link the different sections (A, B, etc.) to an explanation in the caption.

Lines 291-299: please improve the link between these data and the figures. Not very clear.

Line 338: please add more info about these uncertainties.

Line 345: Please modify: This study is the first…

Conclusion: please improve. It is suggested adding the most significant numerical findings obtained from this study. “In general” is not very meaningful, please modify as more appropriate.

References: the ref. n. 7, 34 and 45 have not been cited in the text. Please also check the match between the years of publication reported in the text and in references section.

Please check the following references for possible addition in the MS:

-          Zeng et al (2015) Association of polyfluoroalkyl chemical exposure with serum lipids in children. Science of The Total Environment, 512–513, 364-370. https://doi.org/10.1016/j.scitotenv.2015.01.042.

-          Lin et al (20229 Association between serum per- and polyfluoroalkyl substances and thrombograms in young and middle-aged Taiwanese populations. Ecotoxicol Environ Saf. 236, 113457. doi: 10.1016/j.ecoenv.2022.113457.

Reviewer 2 Report

Dear Authors,

The conducted analysis of the collected data responds to the current need to analyze the impact of food packaging on the product and consumer health. Many people focus on the quality of food, which can change under the influence of packaging and incorrect storage conditions (temperature).

In addition, the conducted research has a huge impact on the perception of food packaging and may consequently lead to the replacement of certain packaging ingredients with less reactive ones.

The presented analysis of the collected data shows how a given packaging component can affect health and the level of ingredients unfavourable to the body (cholesterol)

The methodology requires a few corrections or explanations, which may be obvious to the authors, but not necessarily to people from outside the scientific circle and dealing with this subject.

Conclusions regarding the work are appropriate and answer the questions posed by the authors,

References are appropriate.

Tables and figures are adequately described, but the figures should be of better quality.

Your manuscript concerns a very important and interesting subject. Additions of compounds to packaging, which have specific functions in their production, can significantly migrate to food and affect the health of the consumer. What such a consumer is not always aware of. Nevertheless, I have a few comments about the content of the article.

1. Section 2.3 on what basis were the parameters selected for the exposure calculated and ananalysed

2. Section 2.4. line 179 Please explain the abbreviation BDM

3. Section 2.5. Please explain the abbreviation MOE

4. Section 3.1. Why were the results converted to the natural logarithm and not, for example, to the decimal one? Is there any justification?

5. It seems to me that you should replace %how much with ‰?

Round 2

Reviewer 1 Report

No further comment.

Congratulations for the excellent work!